# Lead ammunition residues in a hunted Australian grassland bird, the stubble quail (*Coturnix pectoralis*): Implications for human and wildlife health

Jordan O. Hampton[1,2][☯]*, Heath Dunstan[3☯], Simon D. Toop[3‡], Jason S. Flesch[3‡], Alessandro Andreotti[4‡], Deborah J. Pain[5‡]

1 Faculty of Veterinary and Agricultural Sciences, University of Melbourne, Parkville, Victoria, Australia, 2 Harry Butler Institute, Murdoch University, Murdoch, WA, Australia, 3 Game Management Authority, Melbourne, Victoria, Australia, 4 Area Avifauna Migratrice, Istituto Superiore per la Protezione e la Ricerca Ambientale, Ozzano Emilia, Italy, 5 Department of Zoology, University of Cambridge, Cambridge, United Kingdom

☯ These authors contributed equally to this work.
‡ SDT, JSF, AA and DJP also contributed equally to this work.
* Jordan.hampton@unimelb.edu.au

**Data Availability Statement:** All relevant data are within the paper and its Supporting Information files.

## Abstract

Scavenging and predatory wildlife can ingest lead (Pb) from lead-based ammunition and become poisoned when feeding on shot game animals. Humans can similarly be exposed to ammunition-derived lead when consuming wild-shot game animals. Studies have assessed the degree of lead contamination in the carcasses of game animals but this scrutiny has not so far extended to Australia. Stubble quail (*Coturnix pectoralis*) are one of the only native non-waterfowl bird species that can be legally hunted in Australia, where it is commonly hunted with lead shot. The aim of this study was to characterize lead contamination in quail harvested with lead-based ammunition. The frequency, dimensions, and number of lead fragments embedded in carcasses were assessed through use of radiography (X-ray). From these data, the average quantity of lead available to scavenging wildlife was estimated along with potential risks to human consumers. We radiographed 37 stubble quail harvested by hunters using 12-gauge (2.75") shotguns to fire shells containing 28 g (1 oz) of #9 (2 mm or 0.08" diameter) lead shot in western Victoria, Australia, in Autumn 2021. Radiographs revealed that 81% of carcasses contained embedded pellets and/or fragments with an average of 1.62 embedded pellets detected per bird. By excising and weighing a sample of 30 shotgun pellets (all had a mass of 0.75 grain or 48.6 mg), we calculated an average lead load of 78 mg/100 g of body mass. This was a conservative estimate, because fragments were not considered. This level of lead contamination was comparable to hunted bird species examined using similar methods in Europe. The quantity and characteristics of lead ammunition residues found suggest that predatory and scavenging wildlife and some groups of human consumers will be at risk of negative health impacts.

**Funding:** The McKenzie Fellowship program of the University of Melbourne and the Victorian Game Management Authority funded this study. The funders did play a role in the preparation of the manuscript (three employees of the Game Management Authority are authors (HD, SDT and JSF)). The funders did not play a role in the analysis or decision to publish.

**Competing interests:** Three authors (HD, SDT and JSF) are employees of the Game Management Authority, a government agency involved with the regulation of recreational hunting activities in the state of Victoria, Australia. Their employment could not reasonably have interfered with the full and objective presentation of this research. This does not alter our adherence to PLOS ONE policies on sharing data and materials.

## Introduction

Lead (Pb) is a toxic non-essential metal that negatively affects multiple body systems in vertebrates, especially the nervous system. It has not been possible to identify a blood lead level in humans without deleterious effects, and so it is advised that human exposure should be minimised, especially for children [1]. Evidence is accruing that a similar situation exists for other vertebrates [2, 3]. Dietary exposure to lead of ammunition origin poses health risks to humans [4], domestic animals such as hunting dogs [5], scavenging wildlife [6], and wild birds including waterfowl that ingest grit (gastroliths) [7]. Lead can be readily ingested and absorbed from game meat by humans due to fragmentation producing numerous particles too small to be detected and removed during food preparation and mastication [8, 9], as well as cooking processes using acid ingredients (wine, vinegar or marinades) enhancing the solubilization of lead particles [10–12]. Wild birds are particularly affected. Waterbirds, terrestrial gamebirds and other taxa may be poisoned following the ingestion of spent lead ammunition [13–15]. Many species of predatory and scavenging birds [6, 7] and mammals [16, 17] are affected across the world. Lead poisoning is estimated to kill millions of waterbirds every year [5] and suppress the populations of raptors across North America [18, 19] and Europe [20] and thus is of substantial conservation significance.

Given the importance of this issue, many studies have documented levels of lead contamination of wild-shot game meat. These include studies assessing the frequency, dimension, and number of lead fragments embedded in the carcasses of ungulates, waterbirds and 'upland' game birds to evaluate the risk related to the consumption of game meat [8, 9]. There is currently considerable global focus on the health and environmental risks resulting from the use of lead ammunition in hunting, and its replacement with non-toxic alternatives is increasingly being mandated or considered across the world [21], especially in Europe [22], North America [23] and Japan [24]. In Australia, despite the popularity of hunting, there have been few moves to transition to lead-free ammunition, and there have been no data available on lead contamination in game birds hunted on this continent [25].

There are few native non-waterfowl bird species that can be legally hunted in Australia and the only such species that can be hunted in the state of Victoria is the stubble quail (*Coturnix pectoralis*). Stubble quail are a small (~100 g) ground-dwelling galliforms that exhibit nomadic behaviour, being capable of long-distance dispersal [26]. They are commonly harvested using shotguns and gundogs on privately owned agricultural land where cropping is the primary land use (Fig 1), and public lands (game reserves), and have been hunted in this way for over 100 years [27]. Currently, approximately 200,000 stubble quail are harvested each year in Victoria by approximately 28,000 licenced hunters [28], all are destined for human consumption, and most are hunted with lead shot. While lead shot has been banned for waterfowl hunting in many (but not all) Australian jurisdictions, its use remains legal for non-waterbirds including grassland birds [25], notably the stubble quail. A 2020 survey of Victorian game licence holders revealed that an estimated 81% (95% CI = 76–84%) of stubble quail hunters still use lead shot [29].

As all harvested and retrieved stubble quail are consumed by humans, the continued use of lead gunshot may present a health risk to consumers, especially frequent consumers and groups particularly vulnerable to the effects of lead, including children. An unknown number of shot and unretrieved stubble quail is also eaten by scavenging or predatory raptors, with potential negative effects on their population dynamics [30]. In spite of the long-term popularity of hunting stubble quail in southeastern Australia, no data relating to lead contamination have previously been available for this species. Here, we assess lead contamination of the carcasses of stubble quail harvested by recreational hunters in Victoria using lead shot. This is the

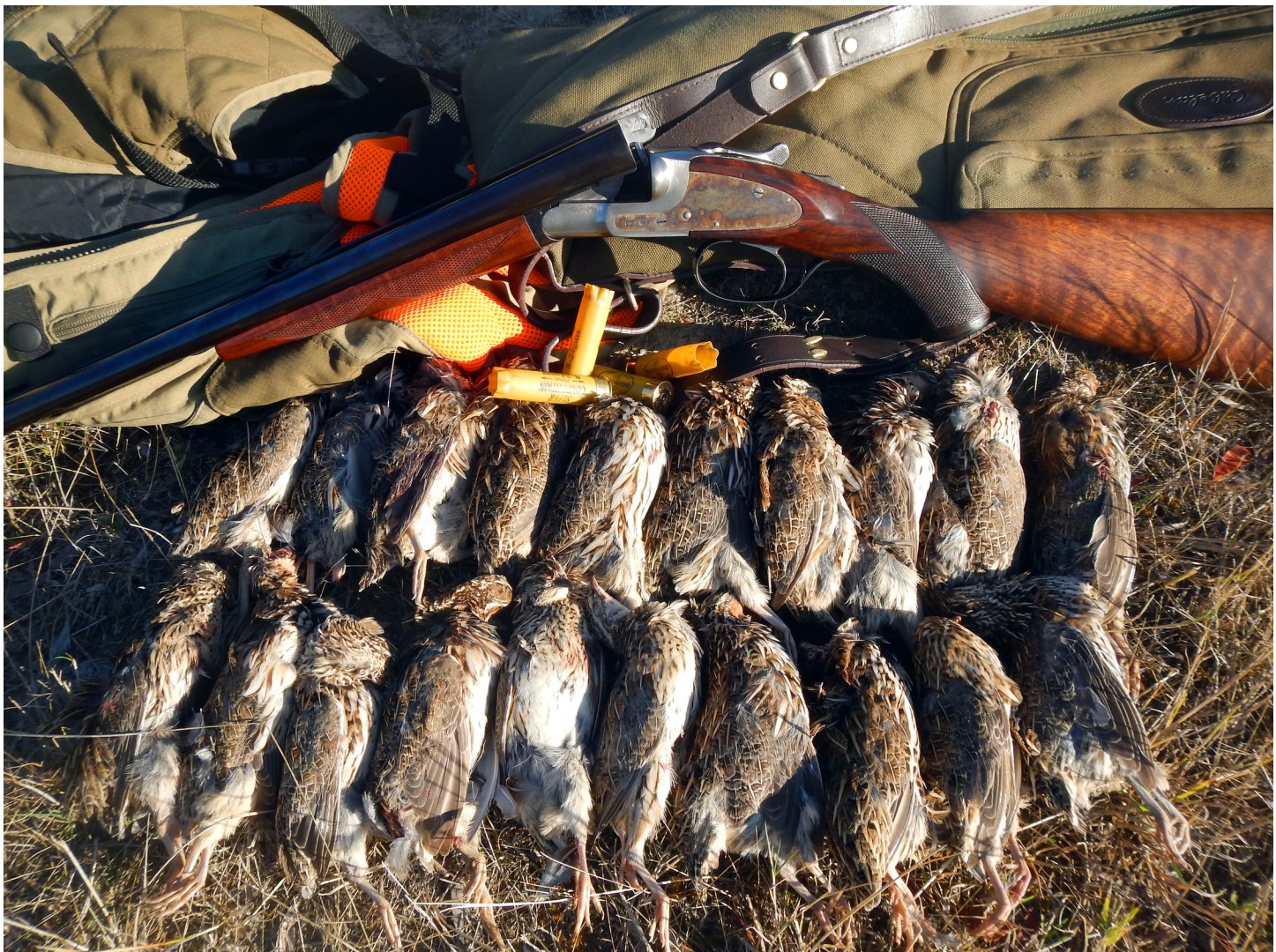

**Fig 1. Typical methods used for recreational hunting of stubble quail (*Coturnix pectoralis*) in south-eastern Australia: Shotgun with lead shot.** This image shows the size of the birds relative to a firearm and a typical harvest for one hunter in one day: 20 birds. Photo: Heath Dunstan (HD).

first description of lead contamination of Australian grassland game birds harvested with lead-based ammunition.

## Materials and methods

### Hunting methods

To produce results comparable to those of published international studies, we largely followed the methods employed in published studies of European game birds [31, 32]. Our study used donated hunter-collected birds harvested with #9 lead shot (2.0 mm (0.08") diameter), from 28 g (1 oz) cartridges fired from 12-gauge (2.75") shotguns on private property in western Victoria, Australia (Fig 1) in Autumn (April–June: the legally allowed hunting season), 2021 [33]. Donated birds were collected by two hunters operating at two sites over six days. Hunting with shotguns is the only technique permitted for recreational hunting of stubble quail in Victoria [29]. No birds were expressly killed for this study.

## Radiography

Harvested birds were immediately labelled with unique numbered metal tags (Fig 2), placed in zip-lock plastic bags and frozen at -20˚C. Birds were thawed and radiographed in a lateral position using a portable veterinary radiography unit (Cuattro, Golden, US) [34] set at 68 kVp and 1.0 mAs. From radiographs, the number of embedded pellets, their anatomical distribution, and the number and size of fragments [35] and their anatomical distribution were recorded (Fig 2).

To describe the distribution of lead pellets and fragments, we subdivided the body of each bird into seven sectors so that the anatomical parts normally consumed by humans or scavenging wildlife could be easily treated separately: 1) head and neck, 2) wings, 3) humerus and pectoral girdle, 4) thorax, 5) abdomen, 6) femur and tibiotarsus, and 7) tarsus and metatarsus [31]. We regarded Sectors 3, 4, 5, and 6 as corresponding to edible parts, i.e., those most commonly eaten by humans and scavenging wildlife.

From radiographs, we counted the number of whole shot pellets embedded in each sector. Thereafter, we assessed the frequency of lead fragmentation, recording the number of 'fragmentation centers' (clusters of radiodense particles or single macrofragments >0.5 mm) [31]. 'Microfragments' (<0.5 mm) are not generally easily detectable to the naked eye on radiographs [36, 37], although this can vary with image resolution. The fragments were scored independently of their dimensions as follows: 0 = none visible; 1 = 1–2 macrofragments; 2 = 2–4 microfragments; and 3 = >4 fragments (regardless of their size). The average level of lead contamination (mg of lead/100 g of body weight) was then calculated [31].

## Dissection

To confirm shot type, evaluate the quantity of the embedded lead and the proportion of the pellet mass fragmented into small particles, we excised 30 shotgun pellets from dissected shot quail. These pellets were washed, dried, and accurately weighed by means of a Hornady® G3-1500 electronic scale (accuracy = ± 0.1 grain (6.5 mg)) (Hornady, Grand Island, US). Shot type was checked with the donating hunters and confirmed by inspection (magnetic, visual and

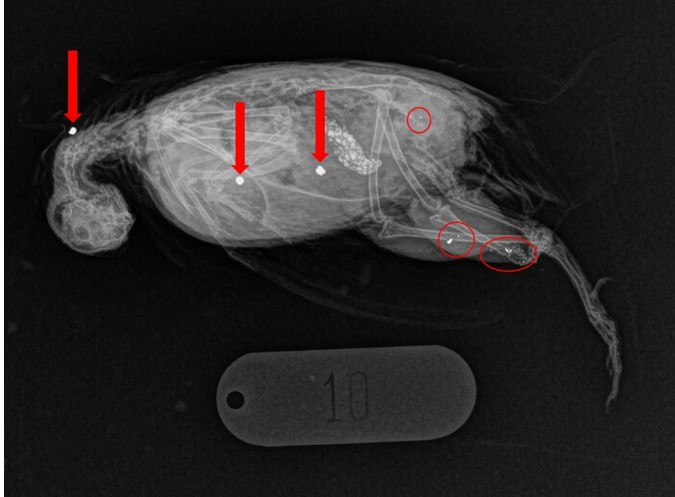

**Fig 2. Embedded lead shot (large bright white objects) and fragmentation centers (red circles) detected via radiography in a harvested stubble quail (*Coturnix pectoralis*): Three pellets and three fragmentation centers are visible in this bird.**

malleability testing) of dissected embedded shot. This comprised: 1) testing whether each pellet was attracted to a magnet, 2) a description of surface colour, and 3) testing of deformability by hand, using a pair of pliers [38, 39].

### Statistics

Descriptive statistics were used to assess the frequency and distribution of lead pellets and fragments in examined stubble quail.

An observational Animal Ethics Committee (AEC) licence from Murdoch University (O3103/19) supported this study.

## Results

We examined 37 stubble quail. The mean weight of the examined stubble quail was 101 g (SD = 12, range = 74–122, $n$ = 37). Lead residues were found in the majority of carcasses: radiographs revealed that 30 birds (81.1%) contained either embedded pellets and/or fragments and 28 birds (75.7%) contained these lead residues in the anatomical regions most commonly eaten by scavenging wildlife or humans. In 14 birds (37.8%), we detected both whole pellets and lead fragments; in 10 (27.0%) only pellets; in 6 (16.2%) we found only small fragments; and in 7 (18.9%) we found neither pellets nor fragments. Pellets and fragments were embedded in all body sectors.

### Shotgun pellets

Radiographs (Fig 2) revealed 60 pellets in 24 stubble quail (mean = 1.62, SD = 1.67, range = 0–5, $n$ = 37). Most of the pellets (65.6%; $n$ = 51) were located in the edible sectors (3–6) with the remaining 9 pellets (15%) found in the head/neck, with no pellets detected in the wings or distal parts of legs. All shotgun pellets excised from dissected birds ($n$ = 30) had a mass of 0.75 grains (48.6 mg). All excised pellets were non-magnetized, dark grey in colour and sufficiently soft to be deformed by pliers, confirming that they were lead [9, 38, 39] and consistent with the statements of donating hunters.

### Fragments

A minimum of 38 fragmentation centers (Fig 2) were detected in 26 stubble quail (mean = 1.0, SD = 1.3, range = 0–5, $n$ = 37). Most of the fragmentation centers ($n$ = 25; 65.8%) were located in the edible sectors (3–6) with smaller fractions found in the distal parts of legs ($n$ = 6; 15.8%), wings ($n$ = 5; 13.2%), and head/neck ($n$ = 2; 5.3%). (Table 1). In 19 cases (50%) of the fragments were assigned to Score 3 (>4 fragments), in 10 (26.3%) were Score 2 and in 9 (23.7%) were Score 1. Some published studies of lead contamination in small mammalian species such as Belding's ground squirrels (*Spermophilus beldingi*) have performed validation studies for radiographic fragment data by dissolving and digesting whole carcasses and manually

**Table 1. Distribution of lead fragment scores in different body sectors.**

| Score | Head / neck | Wings | Humerus / pectoral girdle | Thorax | Abdomen | Femur / tibiotarsus | Tarsus / metatarsus | TOTAL |
|---|---|---|---|---|---|---|---|---|
| Score 1 | 1 | 0 | 2 | 0 | 2 | 2 | 2 | 9 |
| Score 2 | 0 | 2 | 2 | 0 | 2 | 2 | 2 | 10 |
| Score 3 | 1 | 3 | 4 | 2 | 2 | 5 | 2 | 19 |

Fragment scores: 0 = none visible; 1 = 1–2 macrofragments; 2 = 2–4 microfragments; 3 = >4 fragments [31].

extracting bullet fragments [35, 40]. We did not perform any validation via whole carcass digestion so our fragment number data are likely to be underestimates [35].

### Lead load

We calculated an average number of embedded pellets per unit of body mass (1.61/100 g of body weight) and resultant lead load of 78.2 mg/100 g of body mass. This estimate can be regarded as conservative because fragments were not considered. This lead mass was compared to equivalent estimates for hunted bird species examined using similar methods in Europe and Greenland (Table 2). Where studies cited in Table 2 reported the shot size (e.g. #9), but not the corresponding shot mass (e.g. 48.6 mg), we used standardized ballistic tables for shot metrics [41] to calculate shot mass.

## Discussion

To our knowledge, this is the first study to report embedded lead contamination levels in a harvested Australian game bird. Lead contamination was considerable, with 81% of birds contained embedded pellets and/or fragments and an average lead load of 78 mg/100 g of body mass. The embedded lead load in harvested stubble quail was comparable to most European game bird species studied using similar methods (Table 2). However, there was considerable variation, with stubble quail having lead loads that were 289% those reported for European starlings (*Sturnus vulgaris*) [31] but only 68% of those reported for common eider (*Somateria mollissima*) [44, 46]. Our results were also broadly similar to results from oven-ready game-birds of 6 species from the UK, where radiography showed that 65% of birds contained at least one large fragment (estimated to be at least the size of half a pellet) [9].

### Risks to human health

The weight of a game meal is generally considered to range from about 100–200 g of meat per adult [4, 9, 44, 48]. Assuming that about half of the weight of a stubble quail is edible meat, a single portion for an adult consumer would comprise approximately 2–4 oven-ready stubble quail. Our study reveals a high probability that at least one bird in a meal could be contaminated by whole pellets or visible fragments; more are likely to contain tiny fragments. Most whole pellets or large fragments will probably be removed during food preparation or at the table, although occasionally pellets may be inadvertently ingested and some associated lead solubilized. Due to the small surface area to volume ratio of lead pellets and large fragments, only a small proportion of the lead mass is likely to be absorbed into the bloodstream. Rare exceptions include cases where lead shot or bullets are retained in the intestinal tract resulting in elevated blood lead and/or appendicitis [49–51]. However, two factors mean that the risks are higher than this would otherwise imply. These are the presence of very small fragments that cannot readily be removed during food preparation [52], and the solubilization of lead from pellets during the cooking process [10–12].

During the passage of shot through gamebirds, many micro-fragments are widely distributed throughout the flesh and tissue lead concentrations are frequently highly elevated. High lead concentrations have been found in all meals prepared from gamebirds found by X-ray to contain many lead shot, and also in some meals prepared from birds with few or no shot visible on X-rays [9]. This probably results from the presence of lead tiny fragments left embedded in tissue when shot passes through the body. The coarse resolution of two-dimensional radiographic studies makes it difficult to identify lead fragments smaller than about 0.1 mm [4]. However, many lead particles in animals shot with lead ammunition may be considerably smaller than this [52]. The implication is that, even if whole shot and large lead fragments in

**Table 2. Lead contamination in stubble quail (*Coturnix pectoralis*) from south-eastern Australia in comparison to published results from European game bird species.** This table is updated from a past study [31]. Mean body mass and pellet sizes were taken from standard ornithological texts and from hunting sources, respectively.

| Game bird species | *n* birds | *n* pellets/ bird | Reference for *n* of shot | Body mass (g) | Reference for mass | *n* pellets/ 100 g | Shot size | Shot mass per pellet (mg) | Pb mass (mg)/ 100 g |
|---|---|---|---|---|---|---|---|---|---|
| Stubble quail (*Coturnix pectoralis*) | 37 | 1.62 | This study | 101 | This study | 1.61 | 9 | 48.6 | 78 |
| European starlings (*Sturnus vulgaris*) | 196 | 0.65 | [31] | 70 | [42] | 0.93 | 10–11 | 29.1 | 27 |
| Red-legged partridge (*Alectoris rufa*) | 64 | 3.67 | [11] | 500 | [43] | 0.73 | 5–6 | 130.1 | 95 |
| Thick-billed murre (*Uria lomvia*) | 50 | 3.7 | [44] | 900 | [45] | 0.41 | 4 | 175.6 | 72 |
| Common eider (*Somateria mollissima*) | 25 | 10.40 | [46] | 2000 | [47] | 0.52 | 0–6 | 219.2 | 114 |
| Eurasian woodcock (*Scolopax rusticola*) | 59 | 3.64 | [32] | 304 | [32] | 1.20 | 7.5–12 | 37.3 | 45–52 |

stubble quail and other game species are removed at the table, numerous tiny lead fragments are likely to remain in the majority of birds. These are likely to be more readily solubilized and absorbed than whole shot due to their larger surface area/volume ratio [11].

Collectively these results suggest that the availability of lead from ammunition particles is strongly related to their surface area and not just their mass, since the larger surface areas of small particles facilitates lead solubilization during cooking or digestion [9]. In addition to fragmentation, certain game storage and cooking methods may also influence the availability of lead in game meat meals [10–12]. Consequently, health risks from consuming stubble quail will be related to lead concentrations in the meat after removal of whole shot and large fragments, and the frequency of consumption of stubble quail over time.

One UK study found lead concentrations in edible gamebird tissue, after the removal of shot and large fragments, to be on average 1.181 ppm (wet weight) for six species of gamebird (*n* = 121), with no significant variation in tissue lead concentration across the species [9] (Table 5). This mean level is approximately twelve times the Maximum Levels (MLs) permitted in domesticated meat (cattle, sheep, pigs and poultry) marketed in the EU (EC 2006) and listed in the Codex Alimentarius General Standard for Contaminants and Toxins (2019) for commodities moving in international trade [53]. While MLs for lead have not so far been set specifically for game meat within global standards or EU law, it has been recommended that such standards be set [54] and the profile of this issue is increasing internationally.

Although, as discussed, a greater mass of lead gunshot relative to body weight is generally found in large gamebird species rather than in small species like the stubble quail (Table 2) [9], this does not necessarily translate to the lead concentrations found in meat consumed by humans [9]. This is because tiny fragments left by the passage of lead projectiles may be responsible for much of the final lead loads to which human consumers will be exposed. In six UK gamebird species, while body size was positively related to the number of retained shot, lead concentrations in realistic gamebird meals did not vary significantly between species [9].

Therefore, given both the frequency of whole pellets and fragments recorded in stubble quail it can be inferred that the people who eat stubble quail in Australia are likely to be comparably exposed to lead to those consuming birds shot with lead ammunition in Europe and Greenland, including partridges, seabirds and songbirds. The elevated lead levels found in gamebirds shot with lead ammunition are of concern for the health of human consumers, especially those who consume such meat frequently, or are especially vulnerable to the effects of lead, such as children and pregnant women [4]. Additional work analysing lead

concentrations in the edible meat of stubble quail following the removal of readily visible ammunition particles, to simulate realistic human exposure, would further confirm this.

## Risks to scavenging and predatory birds

In addition to human health risks, considerable risks exists for scavenging and predatory raptors that feed on shot game species that are wounded or killed but not recovered by the hunter [7]. The lead loads in Table 2 are more directly relevant to scavengers and predators than humans. In contrast to human consumers, scavenging and predatory wildlife will ingest whole shotgun pellets along with fragments [55]. Raptors ingesting lead shotgun pellets may eliminate them rapidly through defecation or regurgitation, or retain them for days or weeks and their highly acidic stomach conditions can result in considerable erosion/solubilization of ingested shot [56]. Consequently, the lead loads from shot found in stubble quail and other shot species (Table 2) along with the numerous lead fragments, together present considerable risks. Among raptors preying on stubble quail, the wedge-tailed eagle (*Aquila audax*) is known to accumulate harmful amounts of lead in bone as a result of the ingestion of lead-based ammunition [57–59]. Given that this eagle species is a K-strategist, with a high adult survival rate and a low productivity [59, 60], lead poisoning likely exerts adverse effects on it at the population level, as observed in other long-lived apex avian predators with similar life history traits [18, 20].

We conclude that the lead load in stubble quail harvested with lead shot in Australia is similar to that for game bird species harvested in a similar manner in Europe and Greenland. The quantity and characteristics of ammunition residues that were found suggest that game meat consumers, including humans and scavenging and predatory birds and other wildlife, are exposed to considerable quantities of lead. The use of lead shot for stubble quail hunting presents avoidable health risks to both people and wildlife which could be eliminated through a transition to lead-free shot as is increasingly happening in other jurisdictions and has already occurred in Victoria for game ducks.

## Supporting information

**S1 Data.**
(CSV)

## Acknowledgments

We thank Daniel Airo-Farulla for assistance with sample collection. We acknowledge the Wadawurrung people, the traditional custodians of the lands on which this work was conducted.

## Author Contributions

**Conceptualization:** Jordan O. Hampton, Heath Dunstan.

**Data curation:** Jordan O. Hampton.

**Formal analysis:** Jordan O. Hampton, Alessandro Andreotti, Deborah J. Pain.

**Funding acquisition:** Jordan O. Hampton, Simon D. Toop.

**Investigation:** Jordan O. Hampton, Heath Dunstan.

**Methodology:** Jordan O. Hampton, Alessandro Andreotti, Deborah J. Pain.

**Project administration:** Jordan O. Hampton.

**Resources:** Jordan O. Hampton, Simon D. Toop.

**Software:** Jordan O. Hampton.

**Supervision:** Jordan O. Hampton.

**Validation:** Jordan O. Hampton, Alessandro Andreotti.

**Visualization:** Jordan O. Hampton, Heath Dunstan, Alessandro Andreotti.

**Writing – original draft:** Jordan O. Hampton.

**Writing – review & editing:** Jordan O. Hampton, Heath Dunstan, Simon D. Toop, Jason S. Flesch, Alessandro Andreotti, Deborah J. Pain.

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
