## [Decision Letter · Decision Letter 0]

17 Jan 2022

PONE-D-21-40122Lead ammunition residues in a hunted Australian grassland bird, the stubble quail (Coturnix pectoralis): implications for human and wildlife healthPLOS ONE

Dear Dr. Hampton,

Thank you for submitting your manuscript to PLOS ONE. After careful consideration, we feel that it has merit but does not fully meet PLOS ONE’s publication criteria as it currently stands. Therefore, we invite you to submit a revised version of the manuscript that addresses the points raised during the review process.

Be sure to fully address the comments of the reviewers, especially Review 1 who had very constructive feedback.  In particularReviewer 1's concern that your findings are overstated with respect to implications for human health.recommendations by Reviewer 1 for reducing text in the introduction and elsewhere to improve the overall readability of the manuscript.

We look forward to receiving your revised manuscript.

Kind regards,

Myra E Finkelstein

Academic Editor

PLOS ONE

Journal Requirements:

"Three authors (HD, SDT and JSF) are employees of the Game Management Authority, a government agency involved with the regulation of recreational hunting activities in the state of Victoria, Australia. Their employment could not reasonably have interfered with the full and objective presentation of this research."

We note that you received funding from a commercial source: Game Management Authority

4. We note that Figures 1 and 2 in your submission contain copyrighted images. All PLOS content is published under the Creative Commons Attribution License (CC BY 4.0), which means that the manuscript, images, and Supporting Information files will be freely available online, and any third party is permitted to access, download, copy, distribute, and use these materials in any way, even commercially, with proper attribution. For more information, see our copyright guidelines: http://journals.plos.org/plosone/s/licenses-and-copyright.

a. You may seek permission from the original copyright holder of Figures 1 and 2 to publish the content specifically under the CC BY 4.0 license. 

Reviewers' comments:

Reviewer's Responses to Questions

**Comments to the Author**

1. Is the manuscript technically sound, and do the data support the conclusions?

Reviewer #1: Partly

Reviewer #2: Yes

2. Has the statistical analysis been performed appropriately and rigorously? 

Reviewer #1: I Don't Know

Reviewer #2: Yes

3. Have the authors made all data underlying the findings in their manuscript fully available?

Reviewer #1: Yes

Reviewer #2: Yes

4. Is the manuscript presented in an intelligible fashion and written in standard English?

Reviewer #1: Yes

Reviewer #2: Yes

5. Review Comments to the Author

Reviewer #1: Lead exposure to both wildlife and humans is a global concern and I applaud your efforts to better understand the risk associated with shot game birds in Australia. In general I found this manuscript to be well written. However, I do worry that the human health risk implications are based on a sample of 37 birds. Without replication (i.e., samples collected at multiply sites) I wonder if these results are reflective of all stubble quail shooting. Having seem how easily Pb science is discredited here in the US because of relatively minor interpretation issues, I believe studies that are by design meant to be used for management purposes need to be rigorous. I have attached a marked up PDF of you manuscript with comments and questions that hopefully will be helpful in the revision process.

Reviewer #2: This paper fills a gap in reporting of lead exposure risk from upland game hunting in Australia. The authors’ conclusions are succinct and well-supported by the data, and they make a strong case for increased scrutiny of the practices and policies around this wildlife and human health issue. I provide few minor suggestions to be addressed during the final editing process before publication:

L198-199: Suggest changing “Class” to “score for clarity. Referred to as score in the methods, results, and tables up to this point.

L204: Suggest changing to “The average lead mass in stubble quail was similar to most hunted bird species…”

6. PLOS authors have the option to publish the peer review history of their article (what does this mean?). If published, this will include your full peer review and any attached files.

Reviewer #1: No

Reviewer #2: **Yes: **Zeka Glucs

---

## [Author Response · Author response to Decision Letter 0]

1 Mar 2022

Dr Myra E Finkelstein

Academic Editor

PLOS ONE

1 March 2022

Dear Dr Myra E Finkelstein,

Re: Revision of Article (MS# PONE-D-21-40122)

Lead ammunition residues in a hunted Australian grassland bird, the stubble quail (Coturnix pectoralis): implications for human and wildlife health’

Thank you for your letter of 18 January 2022 regarding our manuscript. 

We appreciate the time taken by you and the two reviewers to read and provide comments on our manuscript. We believe that the revised manuscript is substantially improved after addressing those comments. 

Below we detail how we have addressed the editor’s and reviewer’s comments. Changes to the text of the main manuscript are in track changes. The changes we have made are those that we considered most appropriate to address the issues raised but not unduly increase the length of the manuscript. 

Responses (bold) to comments of the reviewers (plain font): 

Editor's Comments to Author:

PONE-D-21-40122

Lead ammunition residues in a hunted Australian grassland bird, the stubble quail (Coturnix pectoralis): implications for human and wildlife health

PLOS ONE

Dear Dr. Hampton,

Thank you for submitting your manuscript to PLOS ONE. After careful consideration, we feel that it has merit but does not fully meet PLOS ONE’s publication criteria as it currently stands. Therefore, we invite you to submit a revised version of the manuscript that addresses the points raised during the review process.

Be sure to fully address the comments of the reviewers, especially Review 1 who had very constructive feedback. In particular:

• Reviewer 1's concern that your findings are overstated with respect to implications for human health.

We found these comments to be reasonable and constructive and have comprehensively revised this section of the manuscript in line with Reviewer 1’s suggestions. 

• Recommendations by Reviewer 1 for reducing text in the introduction and elsewhere to improve the overall readability of the manuscript.

We agree with Reviewer 1 regarding this point. We have made substantial reductions to text length in the identified sections and feel that this has indeed improved the readability of the manuscript. 

We look forward to receiving your revised manuscript.

Kind regards,

Myra E Finkelstein

Academic Editor

PLOS ONE

Journal Requirements:

We have read PLOS ONE's style requirements carefully and we are now confident that our revised manuscript meets them.

"Three authors (HD, SDT and JSF) are employees of the Game Management Authority, a government agency involved with the regulation of recreational hunting activities in the state of Victoria, Australia. Their employment could not reasonably have interfered with the full and objective presentation of this research."

We note that you received funding from a commercial source: Game Management Authority.

Within this Competing Interests Statement, please confirm that this does not alter your adherence to all PLOS ONE policies on sharing data and materials by including the following statement: "This does not alter our adherence to PLOS ONE policies on sharing data and materials.” (as detailed online in our guide for authors http://journals.plos.org/plosone/s/competing-interests). 

We respectful state that the Game Management Authority is not a commercial source. They are a government statutory authority within the government of the state of Victoria, Australia: https://www.gma.vic.gov.au/. As such, they have no commercial interests. Nonetheless, we have not amended our Competing Interests Statement to include the statement “This does not alter our adherence to PLOS ONE policies on sharing data and materials”, as requested. Please find our amended statement at the bottom of this letter.

If there are restrictions on sharing of data and/or materials, please state these. Please note that we cannot proceed with consideration of your article until this information has been declared. 

There are no restrictions on sharing of our data and/or materials.

We have made no amendments to our Competing Interests Statement.

We have moved our ethics statement to the Methods section of our manuscript and deleted it from any other section (Acknowledgments).

4. We note that Figures 1 and 2 in your submission contain copyrighted images. All PLOS content is published under the Creative Commons Attribution License (CC BY 4.0), which means that the manuscript, images, and Supporting Information files will be freely available online, and any third party is permitted to access, download, copy, distribute, and use these materials in any way, even commercially, with proper attribution. For more information, see our copyright guidelines: http://journals.plos.org/plosone/s/licenses-and-copyright.

Our understanding is that neither of Figures 1 nor 2 are copyrighted images. Both photos were taken by authors of this manuscript and have been acknowledged as such. Nonetheless, we have submitted written permission forms from both copyright holders to publish these figures specifically under the CC BY 4.0 license.

a. You may seek permission from the original copyright holder of Figures 1 and 2 to publish the content specifically under the CC BY 4.0 license. 

To repeat our above statement, we believe that neither of Figures 1 nor 2 are copyrighted images. Both photos were taken by authors of this manuscript and have been acknowledged as such. Nonetheless, we have included Content Permission Forms, as requested.

Reviewers' comments:

Reviewer's Responses to Questions

Comments to the Author

1. Is the manuscript technically sound, and do the data support the conclusions?

Reviewer #1: Partly

Reviewer #2: Yes

2. Has the statistical analysis been performed appropriately and rigorously? 

Reviewer #1: I Don't Know

Reviewer #2: Yes

3. Have the authors made all data underlying the findings in their manuscript fully available?

Reviewer #1: Yes

Reviewer #2: Yes

4. Is the manuscript presented in an intelligible fashion and written in standard English?

Reviewer #1: Yes

Reviewer #2: Yes

5. Review Comments to the Author

Reviewer #1 

Lead exposure to both wildlife and humans is a global concern and I applaud your efforts to better understand the risk associated with shot game birds in Australia. In general I found this manuscript to be well written. However, I do worry that the human health risk implications are based on a sample of 37 birds. Without replication (i.e., samples collected at multiply sites) I wonder if these results are reflective of all stubble quail shooting. Having seem how easily Pb science is discredited here in the US because of relatively minor interpretation issues, I believe studies that are by design meant to be used for management purposes need to be rigorous. I have attached a marked up PDF of you manuscript with comments and questions that hopefully will be helpful in the revision process.

L 26: Try to be as specific as possible, I would phrase this as lead-based ammunition, not all ammunition contains Pb.

Re-worded as suggested. 

L 32: Is this true for all of Australia of just Victoria? Seems odd to just focus on Victoria.

Australian hunting laws vary by jurisdiction (state and territory). We have modified this sentence to give a broader perspective: “Stubble quail (Coturnix pectoralis) are one of the only native non-waterfowl bird species that can be legally hunted in Australia, where it is commonly hunted with lead shot.” (new lines 31–33).

L33: Harvested with Pb-based ammunition.

Re-worded as suggested. 

L37: Harvested would be a nicer way to state this.

Re-worded as suggested. 

L37: Folks that do not know much about hunting will not know that a #9 shell (2 3/4 inch) holds 28 gr of pellets, so maybe say of 28 gr of shot and also covert and report that in that to grams.

Re-worded as suggested. 

L39: I would reduce this to just the 65% of harvested birds had Pb fragments/pellets, and the average number of pellets/carcass....keep it concise and focused here.

Reduced as suggested. 

L 43: Why is this approximate, you excised them so you probably weighed them?

We did weigh them. We gave an approximation here as there was minor variation due to mass loss from fragmentation but this was not significant. Re-phrased as suggested. 

L44: Is this because you did not excise those or develop a model to predict their size and mass. These values would be important because smaller fragments are digested more easily in the highly acidic GI system of scavengers and raptors...and thus likely influence Pb exposure more so that larger pellets.

We agree that fragments are biologically important. We did not include them in calculating lead loads for simplicity and to ensure comparability with past published studies. Furthermore, small fragments contribute to the overall lead load (in mg) to a very limited extent.

L51: For a short manuscript like this (i.e., limited data) I would encourage you to shorten this Intro considerable. Currently it is 8 paragraphs long, 4 is more reasonable. Given you don't really do any human risk research (it's anecdotal based on your results), I much of those details on human risk could be removed from the Intro.

L 203: Given the data in table 2 I would actually say it's fairly different, for instance it's ~3-fold higher than the starlings, ~30-40% lower than eiders, etc.

We agree and have shortened the Introduction as suggested. It is now four paragraphs. 

L 53: I see this statement written regularly but I don't agree with it because we can measure Pb concentrations at such low values using ICPMS or other analytical methods. Given in wildlife (birds mostly) we do not see effects of Pb exposure at these exceptionally low concentrations it makes this statement questionable.

We respectfully disagree with the reviewer here. What the reviewer asserts does not appear to be in line with the position of global health authorities on lead poisoning, nor the peer-reviewed literature. Global health authorities agree that there is no evidence for a minimum blood lead threshold below which negative health effects do not occur – indeed in children the evidence suggests that the relationship between blood lead and IQ is not linear and proportionately greater effects are seen at lower blood lead concentrations. For example, the US Centers for Disease Control and Prevention (CDC) recognize that no safe blood lead level in children has been identified and that even low levels of lead in blood have been shown to reduce a child’s learning capacity, ability to pay attention, and academic achievement [1]. They therefore reduced the blood lead reference value (BLRV) to 3.5 µg/dL in October 2021 to identify children with higher blood lead levels so that action can be taken to reduce levels further. The terminology ‘threshold’ is no longer used because no threshold for lead’s effects has been identified. Also the Provisional Tolerable Weekly Intake (PTWI) of dietary lead in humans was scrapped a decade ago as there is no intake level considered to be safe – instead health agencies use benchmark dose-response relationships to measure health effects considered to be significant at a population level – the benchmark dose limit BMDL01 is 1.2 µg/dL in blood for a 1% (1 point) reduction in IQ in children.

As with humans, an increasing number of effects are being documented in other vertebrates including birds at very low levels of exposure and with ever lower blood lead concentrations. This has only become possible because major sources of exposure to lead (e.g. as a petrol additive) have now been massively reduced. We are not aware of any evidence to suggest that the situation in humans and other vertebrates is dissimilar, and behavioural studies are finding effects at very low blood lead concentrations in wild birds, despite the technical complexity of these studies (e.g. [2]). As such, we argue to retain the statement in question. We have added two citations to this sentence to emphasise the nature of this evidence but have otherwise left the statement unchanged (new lines 56–58). 

L59: While this might be true from the perspective of the Pb ban for waterfowl hunting in the US and Canada. Currently there is almost no forward motion on regulating Pb-based ammunition for big game or varmint shooting....California being the only exception.

We agree but point out that this statement asserts only that replacement with lead-free ammunition is being considered. There is substantial forward motion on this issue, especially across Europe and in Japan. This has certainly been the case in many jurisdictions in North America, with previous bans on Federal land in the US and many state-led voluntary transition programs [3]. In Europe, the European Union and the UK are currently considering a complete ban (regulatory restriction) on the use of lead ammunition (gunshot and bullets) under REACH regulations [4]. Many voluntary restrictions are already in place on lead bullet use in the UK and parts of Europe, and supermarkets and game dealers are increasingly refusing to sell game shot with any kind of lead ammunition. Japan is also in the process of banning lead bullets by 2030 [5]. 

L 66: Are you implying juveniles that are exposed to Pb at a young age then suffer reproductive issues? Is this really supported by the literature?

This sentence has been deleted to reduce the length of the Introduction.

L77: This should probably have some citations to support it.

This sentence has been deleted to reduce the length of the Introduction.

L79: Again this needs citations.

This sentence has been deleted to reduce the length of the Introduction.

L88: I would remove the language that is redundant with the next paragraph and combine it with the next paragraph.

Re-written as suggested. 

L94: Remove ‘relatively’. 

This sentence has been deleted to reduce the length of the Introduction.

L101: These estimates are vastly different, is that a function of the boom-bust pop growth, otherwise if not are these accurate? 

This variation is thought to be a function of the boom-bust population cycles of the species, as described in a recent paper that we have cited [6]. Regardless, this statement has been deleted to reduce the length of the Introduction.

L101: Is there no hunting that occurs on public lands (game reserves)? 

There is some hunting that occurs on public lands (game reserves). We have amended the text here to reflect this.

L108: You provide the 95% CI around the estimate of how many licence holders shoot Pb shot, but then this number is just approximated, so the State not have a more exact estimate with CIs?

That’s correct, it is an estimate based on a survey of 344 quail hunters [7]. The state of Victoria does not currently conduct any formal data collection regarding ammunition type for licensed quail hunters. 

L125: Are you missing the word "donated" in here somewhere, otherwise how did the study acquire birds?

We were missing the word ‘donated’ here. Thank you for picking this up – it has now been corrected. 

L127: I would replace the figure of the harvested quail and a shotgun with one that actually shows the study area location.

We appreciate the suggestion but feel that it is relatively easy for readers to find the location of western Victoria on a map or online but depictions of what quail hunting looks like in the field are much harder to find. No change made. 

L133: Unclear why you mentioning this, again a map of the actual location of research would be more useful here.

This text was added to show the size of stubble quail to any readers unfamiliar with the species. Our reasons for not including a map figure are given above. No change made. 

L138: Were these samples collected from multiple locations (ie., representative of all birds) or where they all collected at one site. I would spell this out, gets to how representative your results are. If you simply collected all birds at one site, one, day, then your ability to draw as much inference as you do is limited.

The samples were collected from two hunters operating at two sites over six days. We have added this detail as suggested and feel that, with this breadth of samples, and the relatively uniformity of quail hunting methods used [7], we are justified in drawing as much inference as we do with the disclaimers we have provided. 

L149: Do you know that all birds were in fact shot with Pb pellets? Could any of these observed pellets in fact be steel shot?

We do know that all birds were shot with lead pellets. This was checked with the donating hunters so we did not think that this required clarification. However, we are happy to provide additional detail as it has been requested. Shot type was confirmed by inspection (magnetic, visual and malleability testing as per [8]) of dissected embedded shot. All were non-magnetized, grey and sufficiently soft to be scratched with a scalpel, suggesting that all were lead and could not have been steel. This information has been added to our Methods (new sub-heading: ‘Dissection’; new lines 194–197) and Results (new lines 221–223) sections. 

L151: is this really accurate for the parts of a bird that is only 100 g to be consumed by scavengers?

This approach seems reasonable to us given that many scavengers active in Australia (e.g. corvids) are relatively small and selective [9]. It is also consistent with the way in which similar published studies have addressed the anatomical distribution of embedded shot [10].

L155: Were these counts ever validated, if not they are most certainly underestimates for fragments (see Pauli and Buskirk 2008 or Herring et al. 2016).

No, we did not validate these results. For whole embedded pellets, our counts were almost certainly accurate, given how obvious these structures are on radiographs and that the maximum number in any one bird was only 5. However, for fragments, we agree that the reviewer makes a good point. We did not have the resources to conduction validation studies as per the suggested studies [11, 12]. Accordingly, we have added an acknowledgment that our fragment count data are likely to be underestimates. 

L167: Normally you would just report the accuracy as +/- 0.1 gr, again I would convert this to grams or mg as well.

Re-worded as suggested. 

L174: I would delete this statement, no need to make excuses for smaller sample sizes.

Deleted as suggested. 

L176: You could be more specific here because you know what they were shot with "Shotgun shell pellets".

Re-worded as suggested. 

L186: It might be useful to calculate what proportion of quail carcasses these represent. You current results simply report 60 pellets in 24 birds, yet the mean is 1.62, suggesting some of those carcasses had a lot of pellets.

We have reported the mean and range for this metric – we feel this is sufficient. No change made. 

L186: I assume this mean is across all birds including those with no pellets.

That is correct. We feel this is clear. No change made.

L189: Why is this an approximate mass, you measured these so report the mean mass and error estimate?

As stated above, this has been re-phrased as suggested. 

L203: Is it possible to include some estimate of error around this estimate? Also you might define how you the average Pb load, assuming you simply multiplied the mean number of pellets (1.62) by the mean mass of the pellets (48.6 mg) you get 78.7 not 78.2 mg.

We haven’t reported any estimate of error here as this is a rate, rather than a mean value. The average lead load was calculated per 100 g as stated, whereas the quail sampled in this study had a mean mass slightly >100 g, accounting for the small difference in calculated values.

L203: Should this be developed only using the pellets that were found in the edible regions, because the pellets in the head/neck etc are not available?

That does sound a reasonable alternative to the approach we have taken, and if we were only focused on human consumers, but we wanted to report the total lead load to allow comparison with past published studies [10], and to allow estimation of the total amount of lead that would be available to any scavenger that ate an entire bird. No change made.

L211: Given the data in table 2 I would actually say it's fairly different, for instance it's ~3-fold higher than the starlings, ~30-40% lower than eiders, etc.

This is a fair point. We have amended this section to recognise this variance and have moved it to the Discussion where we feel it fits better.

L211: Not all of these studies reported what shot size was used/reported by the hunters, subsequently the estimates of mass are inferred?

Shot sizes are standardized, and if made from lead, their mass is known using standard ballistic tables, e.g. https://www.claygame.co.uk/shot-size-info-22 We have added a sentence recognizing that these standard shot tables were the origin of our estimates of shot mass: “Where studies cited in Table 2 reported the shot size (e.g. #9), but not the corresponding shot mass (e.g. 48.6 mg), we used standardized ballistic tables for shot metrics [44] to calculate shot mass.” (new lines 46–49). 

L211: In the second paper the Joahnsen and others published the reported Pb concentrations in murres was 0.73 ug/g and 6.1 ug/g for eiders. The concentrations you present here do not seem to match up nor do they make sense even after accounting for the fact that eiders are ~double the mass of murres.

We understand the confusion here. The paper in question reported lead concentration in meat (as measured via spectroscopic methods), in addition to reporting the lead load as defined as the mass of embedded shot and measured via radiography [13]. These data (e.g. 0.73 ug/g) are not related to the data reported in Table 2. No change made.

L211: Report all these values to the same decimal place.

Changed as suggested. 

L211: This study was redone in the below eider study because the variability in Pb concentrations was too high, you should use the estimates for the murres provided in citation 44, these are the exact same birds

See comment above regarding confusion between lead concentration in meat and lead load in an entire carcass. No change made.

L211: This value is not reported in the cited paper, is it correct?

The mean number of shot pellets in eiders was calculated and reported in the study of Andreotti and Borghesi [10]. Original data were inferred by these authors from Johansen et al. [13], Table 4.

L211: This and the above murre paper do not report these results, where do they come from, if you are just basing this on anecdotal information you should be clear about that because when you report it in this table readers will assume it's from a peer reviewed paper.

These values were originally calculated and reported in the study of Andreotti and Borghesi [10]. These authors reported that “mean body weights and pellet sizes were taken from standard ornithological texts and from hunting sources, respectively” (caption of Table 3).

L221: Those mean number of pellets per birds in the Pain et al. paper are biased in some cases by individual birds with a large number of pellets (e.g., pheasants had the highest mean number of pellets [3.32]) but the mean is biased by two birds, one with 13 pellets and one with 18 pellets. These values probably should have been transformed to normalize the data (e.g., log transformed, then back transform the log mean to get a geometric mean). Your pellet data are not quite as log normally distributed but certainly are not normal.

This is a fair point but we wish to present our data in the simplest manner possible and to maximise comparability with past published studies. We have added the median value of embedded pellets found in our study to provide readers with additional information about the distribution of these data. We have re-written parts of this first paragraph to emphasise the results of own study and have removed discussion of mean shot/ fragment number reported in the published study of Pain et al. [8]. For risk assessment purposes it is generally considered better to use means rather than medians as means represent the average exposure over the course of time – sometimes a bird or a person will be exposed to no pellets, sometimes to 10. This also applies to lead concentrations in meat after pellets and large fragments have been used, and is the approach that has been adopted for lead concentrations in meat by European food safety agencies and in European Union-wide human health risk assessments [14]. No change made.

L222: This is completely speculative and as such does not add any support to previous work. Given your Discussion is fairly long for a relatively limited data set, I would delete this paragraph because it adds little.

We have deleted this paragraph as suggested. 

L241: Why do you need to make any assumptions here, you had the carcasses. Would it not have been more accurate for the human health risk assessment to just weight the meat.

This is a fair point, but we can’t be completely confident as to which portions of quail meat a given hunter may eat. For example, some may eat only the breast meat, and others, breast and leg meat. As such, we couldn’t accurately estimate the mass of each bird that an average hunter would consume. No change made.

L243: This implies the average meal is 160-300 g, yet you cite papers suggesting it's 100-200 g, can you explain this discrepancy.

We agree there is some discrepancy between the average meal size suggested by published international literature and that reported by Australian quail hunters in personal communications. To reconcile this discrepancy, we have removed reference to the personal communication and have deferred to the international estimate, which is likely to be more accurate. 

L271: This is somewhat of an apples to oranges comparison, big game species are typically shot with fragmenting Pb-based bullets, shotgun pellets are not intended to fragment intentionally. Your own results show this, you found XX% of stubble quail carcasses contained Pb fragments.

We understand the point made but, regardless of the intentions of ammunition manufacturers, both lead-based bullets and shot do fragment so the premise explained here is valid. Nonetheless, we have deleted this section to reduce the length of the Discussion, as suggested below.

L277: You have 4 paragraphs that largely focus on micro Pb fragments in game meat....this could/should be limited to 1 paragraph.

We agree that our initial discussion of small lead fragments may have been excessive given the length of our manuscript. This section has been reduced in length as suggested. 

L289: No need to say significant, it's inferred

This sentence has been removed. 

L298: Does this number really come from the cited source? I can't find it.

In Table 5 of Pain et al. [8], values for 6 individual species (WWT) are listed (means and sample sizes). The grand arithmetic mean of these is 1.181. While this is not explicitly stated in the paper, it is simple to calculate from Table 5 [8]. No change made.

312: No need to say significantly, it's inferred when you say positively

Removed as suggested. 

L313: I think you need to join these two parts of this sentence, replace the comma with a "and".

Re-worded as suggested. 

L316: The issue I see with drawing this inference from your results is that you do not know what concentrations of Pb is retained in the meat if you remove the Pb pellets...as you elude to at the end of the paragraph. Having hunted both waterfowl and upland game birds for over 30 years, I can say that I have only detected whole pellets in a small proportion of the birds I have consumed. In most cases I suspect game meat consumers likely do detect and remove whole pellets if they are present in the cooked meat.

We agree with this partly but feel that this is sufficient evidence from numerous published studies globally demonstrating the link between radiographic evidence of lead residues and concentrations of lead measured via spectroscopic methods to live this as is. No change made. Elevated lead concentrations are also found in the meat of game hunted with lead gunshot that has not retained whole shot as a result of fragmentation as it passes through an animal. 

L337: Relevant? unclear what you mean here, reword.

Re-worded as suggested.

L340: Totally speculative, word that as may exert.

We respectfully disagree with the reviewer on this point. Many studies carried out around the world have revealed that eagle, condor and vulture species are very sensitive to lead poisoning and their demography is negatively affected by lead. A spate of high-profile papers published in the past year have reiterated this point [15-17] and three recent papers have confirmed that the species in question (the wedge-tailed eagle Aquila audax) demontrates similar levels of lead exposure to the species for which demographic modelling has been completed [18-20]. We argue to retain this sentence with some changes and additional references added to better support our statement (new lines 394–397). 

L341: This is a fairly bold statement for a study that has a n of 37, with potentially all samples coming from one site.

We recognize that our sample size was small, but not all of our samples came from a single site (not that that would affect lead residues in their tissues from shooting). Nonetheless, we feel that this is a modest statement that is supported by our data. No change made.

Reviewer #2

This paper fills a gap in reporting of lead exposure risk from upland game hunting in Australia. The authors’ conclusions are succinct and well-supported by the data, and they make a strong case for increased scrutiny of the practices and policies around this wildlife and human health issue. I provide few minor suggestions to be addressed during the final editing process before publication:

L198-199: Suggest changing “Class” to “score” for clarity. Referred to as score in the methods, results, and tables up to this point.

Changed throughout as suggested. 

L204: Suggest changing to “The average lead mass in stubble quail was similar to most hunted bird species…”

We have changed this sentence to address the comments of Reviewer 1.

References

1. US Centers for Disease Control and Prevention (CDC). CDC updates blood lead reference value to 3.5 µg/dL. Atlanta, USA: US Centers for Disease Control and Prevention (CDC) 2021.

2. Ecke F, Singh NJ, Arnemo JM, Bignert A, Helander B, Berglund ÅMM, et al. Sublethal lead exposure alters movement behavior in free-ranging golden eagles. Environmental Science & Technology. 2017;51(10):5729–36.

3. Schulz JH, Stanis SAW, Hall DM, Webb EB. Until It's a regulation It's not my fight: Complexities of a voluntary nonlead hunting ammunition program. Journal of Environmental Management. 2021;277:111438.

4. European Commission. Request to the European Chemicals Agency to prepare a restriction proposal on the placing on the market and use of lead in ammunition (gunshot and bullets) and of lead fishing tackle conforming to the requirement of Annex XV to REACH. Brussels, Belgium: European Commission, 2019.

5. Ishii C, Ikenaka Y, Nakayama SM, Kuritani T, Nakagawa M, Saito K, et al. Current situation regarding lead exposure in birds in Japan (2015–2018); lead exposure is still occurring. Journal of Veterinary Medical Science. 2020;82(8):1118–23.

6. Moloney PD, Gormley AM, Toop SD, Flesch JS, Forsyth DM, Ramsey DSL, et al. Contrasting long-term trends in the harvest of native and introduced species by recreational hunters in Australia revealed using Bayesian modelling. Wildlife Research. In press.

7. Game Management Authority. GMA hunting methods survey 2020. Melbourne, Australia: Game Management Authority, 2020.

8. Pain DJ, Cromie RL, Newth J, Brown MJ, Crutcher E, Hardman P, et al. Potential hazard to human health from exposure to fragments of lead bullets and shot in the tissues of game animals. PloS One. 2010;5(4):e10315.

9. Woodford LP, Forsyth DM, Hampton JO. Scavenging birds at risk of ingesting lead bullet fragments from kangaroo and deer carcasses in south-eastern Australia. Australian Field Ornithology. 2020;37:112–6.

10. Andreotti A, Borghesi F. Embedded lead shot in European starlings Sturnus vulgaris: an underestimated hazard for humans and birds of prey. European Journal of Wildlife Research. 2013;59(5):705−12.

11. Pauli JN, Buskirk SW. Recreational shooting of prairie dogs: A portal for lead entering wildlife food chains. The Journal of Wildlife Management. 2007;71:103–8.

12. Herring G, Eagles-Smith CA, Wagner MT. Ground squirrel shooting and potential lead exposure in breeding avian scavengers. PLoS One. 2016;11(12):e0167926.

13. Johansen P, Asmund G, Riget F. High human exposure to lead through consumption of birds hunted with lead shot. Environmental Pollution. 2004;127(1):125–9.

14. Gerofke A, Ulbig E, Martin A, Müller-Graf C, Selhorst T, Gremse C, et al. Lead content in wild game shot with lead or non-lead ammunition–Does “state of the art consumer health protection” require non-lead ammunition? PloS One. 2018;13(7):e0200792.

15. Hanley BJ, Dhondt AA, Forzán MJ, Bunting EM, Pokras MA, Hynes KP, et al. Environmental lead reduces the resilience of bald eagle populations. The Journal of Wildlife Management. 2022;86(2):e22177.

16. Slabe VA, Anderson JT, Millsap BA, Cooper JL, Harmata AR, Restani M, et al. Demographic implications of lead poisoning for eagles across North America. Science. 2022;375(6582):779–82.

17. Green R, Pain D, Krone O. The impact of lead poisoning from ammunition sources on raptor populations in Europe. Science of the Total Environment. 2022;In press.

18. Lohr MT, Hampton JO, Cherriman S, Busetti F, Lohr, C. Completing a worldwide picture: preliminary evidence of lead exposure in a scavenging bird from mainland Australia. Science of The Total Environment. 2020;715, 135913.

19. Pay JM, Katzner TE, Hawkins CE, Koch AJ, Wiersma JM, Brown WE, Mooney NJ, Cameron EZ. High frequency of lead exposure in the population of an endangered Australian top predator, the Tasmanian wedge‐tailed eagle (Aquila audax fleayi). Environmental Toxicology and Chemistry. 2021;40(1):219–30.

20. Hampton JO, Specht AJ, Pay JM, Pokras MA, Bengsen AJ. Portable X-ray fluorescence for bone lead measurements of Australian eagles. Science of The Total Environment. 2021;789:147998.

Amended Competing Interest Statement

Three authors (HD, SDT and JSF) are employees of the Game Management Authority, a government agency involved with the regulation of recreational hunting activities in the state of Victoria, Australia. Their employment could not reasonably have interfered with the full and objective presentation of this research. This does not alter our adherence to PLOS ONE policies on sharing data and materials.

We hope that our resubmission is acceptable. We reiterate that the changes we have made are those that we considered most appropriate to address the issues raised but not unduly increase the length of the manuscript. We are willing to consider any further comments and suggestions that you have about our manuscript. 

Thank you very much for considering our revised manuscript for publication in PLOS ONE.

Sincerely (for the authors),

Jordan O. Hampton

University of Melbourne, Parkville, Vic 3052, Australia 

E-mail: jordan.hampton@unimelb.edu.au

---

## [Decision Letter · Decision Letter 1]

8 Apr 2022

Lead ammunition residues in a hunted Australian grassland bird, the stubble quail (Coturnix pectoralis): implications for human and wildlife health

PONE-D-21-40122R1

Dear Dr. Hampton,

We’re pleased to inform you that your manuscript has been judged scientifically suitable for publication and will be formally accepted for publication once it meets all outstanding technical requirements. 

Although your paper is acceptable as is for publication, I ask you to consider this additional comment from one of the reviewers:  "I ask that you carefully consider what I'm saying here in my response. While the issues is minor it gets to the importance of being factual in science and our credibility. I previously made the point that I did not agree with the statement on line 56 "No levels of exposure are considered to be safe to humans" You response without a doubt illustrates that lead can be detrimental to both humans and wildlife at very low concentrations. My point was simply that we measure lead concentrations far below these concentrations that you reference. For instance you referenced citation 2 and make the point that a blood lead concentration of 1.2 ug/dL has been shown to result in a 1% reduction in IQ in children. In our lab, our detection limit for Pb is 6-fold lower than this. Similarly you reference the Ecke study that found a relationship between changes in flight patterns in golden eagles at 2.5 ug/dL, again we measure blood lead accurately 12.5-fold lower than this concentration. My point is not to cast doubt on the fact that lead can and does impact both humans and wildlife at very low concentrations (and our own research on birds supports this), but simply to point out the fact that analytically the statement is not entirely accurate. Perhaps in time we will be able to show that these exceptionally low concentrations around the detection limit do in fact influence impact humans and wildlife, but to my knowledge current studies do not support this. Thanks for considering my comments.

Kind regards,

Myra E Finkelstein

Academic Editor

PLOS ONE

Additional Editor Comments (optional):

Reviewers' comments:

Reviewer's Responses to Questions

**Comments to the Author**

1. If the authors have adequately addressed your comments raised in a previous round of review and you feel that this manuscript is now acceptable for publication, you may indicate that here to bypass the “Comments to the Author” section, enter your conflict of interest statement in the “Confidential to Editor” section, and submit your "Accept" recommendation.

Reviewer #1: All comments have been addressed

Reviewer #2: All comments have been addressed

2. Is the manuscript technically sound, and do the data support the conclusions?

Reviewer #1: Yes

Reviewer #2: Yes

3. Has the statistical analysis been performed appropriately and rigorously? 

Reviewer #1: Yes

Reviewer #2: Yes

4. Have the authors made all data underlying the findings in their manuscript fully available?

Reviewer #1: Yes

Reviewer #2: Yes

5. Is the manuscript presented in an intelligible fashion and written in standard English?

Reviewer #1: Yes

Reviewer #2: Yes

6. Review Comments to the Author

Reviewer #1: Thank you for taking the time to carefully consider my comments and suggestions on your manuscript. I believe you have appropriately addressed all of my comments except one. I ask that you carefully consider what I'm saying here in my response. While the issues is minor it gets to the importance of being factual in science and our credibility. I previously made the point that I did not agree with the statement on line 56 "No levels of exposure are considered to be safe to humans" You response without a doubt illustrates that lead can be detrimental to both humans and wildlife at very low concentrations. My point was simply that we measure lead concentrations far below these concentrations that you reference. For instance you referenced citation 2 and make the point that a blood lead concentration of 1.2 ug/dL has been shown to result in a 1% reduction in IQ in children. In our lab, our detection limit for Pb is 6-fold lower than this. Similarly you reference the Ecke study that found a relationship between changes in flight patterns in golden eagles at 2.5 ug/dL, again we measure blood lead accurately 12.5-fold lower than this concentration. My point is not to cast doubt on the fact that lead can and does impact both humans and wildlife at very low concentrations (and our own research on birds supports this), but simply to point out the fact that analytically the statement is not entirely accurate. Perhaps in time we will be able to show that these exceptionally low concentrations around the detection limit do in fact influence impact humans and wildlife, but to my knowledge current studies do not support this. Thanks for considering my comments.

Reviewer #2: My comments were not as extensive as my fellow reviewer, but I am satisfied with the authors' responses to my comments/suggestions. I feel the revision has been significantly improved from the original submission both in quality and readability.

7. PLOS authors have the option to publish the peer review history of their article (what does this mean?). If published, this will include your full peer review and any attached files.

Reviewer #1: No

Reviewer #2: No

---

## [Editor Report · Acceptance letter]

12 Apr 2022

PONE-D-21-40122R1 

Lead ammunition residues in a hunted Australian grassland bird, the stubble quail (*Coturnix pectoralis*): implications for human and wildlife health 

Dear Dr. Hampton:

I'm pleased to inform you that your manuscript has been deemed suitable for publication in PLOS ONE. Congratulations! Your manuscript is now with our production department. 

Kind regards, 

on behalf of

Dr. Myra E Finkelstein 

Academic Editor

PLOS ONE